# COVID-19 Fear Impact on Israeli and Maltese Female “Help” Profession Students

**DOI:** 10.3390/ijerph20053968

**Published:** 2023-02-23

**Authors:** Mor Yehudai, Marilyn Clark, Andrew Azzopardi, Shai-li Romem Porat, Adi Dagan, Alexander Reznik, Richard Isralowitz

**Affiliations:** 1Regional Alcohol and Drug Abuse Research Center, Ben Gurion University of the Negev, Beer Sheva 84105, Israel; 2Department of Psychology, Faculty for Social Wellbeing, University of Malta, 2080 Msida, Malta; 3Department of Youth and Community Studies, Faculty for Social Wellbeing, University of Malta, 2080 Msida, Malta

**Keywords:** COVID-19 fear, university students, women, professional students, mental health, substance use, Israel, Malta

## Abstract

Background: The aim of this cross-sectional study was to examine the impact of COVID-19 fear on the well-being of Israeli and Maltese female “help” profession (e.g., social work and psychology) undergraduate students. This cross-national comparison includes factors of depression, anxiety, anger, loneliness, nervousness, substance use, eating behavior, burnout, and resilience. The study hypothesis is that country status, even with different social–cultural characteristics including religiosity, is not a significant factor associated with COVID-19 fear impact on select behavioral characteristics of female university students. Methods: A total of 453 female “help” profession students completed an online survey from January to July 2021. Various statistical methods of analysis including regression were used for this study. Results: The mean COVID-19 fear scores were the same among Israeli and Maltese students. Resilience was found to be higher among Israeli females; burnout was found to be higher among those from Malta. Substance use (i.e., tobacco, alcohol, cannabis, stimulants, or prescription drugs) in the last month was reported by 77.2% of the respondents. No significant differences were found for previous-month substance use based on country status. Regardless of country, respondents who reported more previous-month substance use had higher COVID-19 fear and burnout scores, as well as lower resilience. Due to COVID-19, most respondents (74.3%) reported deterioration of their psycho-emotional well-being in the last month; however, no significant differences were found based on country and religiosity statuses. Furthermore, no significant differences were found for eating behavior changes and weight increase based on country and religiosity statuses. Conclusion: Study findings showed the impact of COVID-19 fear on the well-being of Israeli and Maltese female “help” profession undergraduate students. This study examined only female students; however, additional research is needed to address male students and their experiences. Prevention and treatment intervention measures aimed to increase resilience and decrease burnout, including those that can be made available on campus, should be thought about by university administration personnel and student association leaders in consultation with mental health professionals.

## 1. Introduction

In December 2019, the first COVID-19 cases were reported. From that time on, the pandemic has had a profound impact on health, psychological, and economic conditions worldwide. Studies have found that infectious diseases, such as COVID-19, are associated with psychological distress and mental illness [1]. Such conditions include stress, anxiety, depression, insomnia, anger, fear, stigma [2,3,4], substance misuse [5,6,7], and other psycho-emotional-related issues.

The first COVID-19 case in Malta, an island country in the Mediterranean Sea with slightly more than a half million people, was reported in March 2020. Following a declared public health emergency, the government instituted mitigation measures including isolated work and study practices, social distancing, and limitations on public gatherings [8]. In Israel, with a population of nearly 10 million people, similar policies were initiated to combat COVID-19, including strict quarantine; cancellation of theater performances, movies, and attendance at sporting events; school and university closure; cessation of aviation and railway travel; cancellation of most cross-border movement except for emergency situations; mandatory mask use; and other restrictive conditions [9]. 

In both countries, vaccination was readily available. However, fear of COVID–19 continued to impact young people as the pandemic ran its course. Based on research from both countries at the onset of COVID-19 infection, women were more likely to report increased negative emotions, including those of a mental health nature such as depression and post-traumatic stress disorder [10]. 

Among university students, research shows that the well-being of a significant portion of this population was negatively affected by the pandemic [11]. Furthermore, in response to prevailing conditions, universities in Israel, Malta, and other countries shifted from traditional classroom learning to internet instruction along with social distancing [12]. Such actions were taken despite warnings about a mental health tsunami [13] resulting from social isolation [14], substance misuse [15,16], and other maladaptive coping mechanisms [17,18]. 

Most institutions of higher education were able to respond to the pandemic with remote communication methods. However, online teaching could not fully replace face-to-face learning, and the closure of educational institutions resulted in student isolation affecting the learning process as well as social interaction that contributes to interpersonal support in times of disaster [19,20,21]. 

By virtue of their status in the human development trajectory, many young adults faced unique mental health challenges caused by the pandemic that negatively affected their independence as well as personal and professional attributes [22]. Several studies have elaborated on the psychological and physical health processes negatively affected by COVID-19 fear [5,23,24,25,26,27,28]. The following is a brief review of the impact of COVID-19 on key factors and conditions.

### 1.1. Gender

Numerous studies on the harmful effects of COVID-19 evidence such conditions are not equally distributed based on gender status. Women, especially those who are caregivers, tend to experience more negative consequences [29,30], including those of a mental health nature [31,32]. From an education perspective, particularly in tertiary education, women tend to often excel beyond the levels achieved by their male counterparts [33]. This development contributes to narrowing the gender gap associated with advantage and privilege. However, it has been noted that the impact of the pandemic on women, including factors of social isolation, stress, and diminished mental health, may be a setback for them in terms of their professional development and accomplishment [34,35]. For example, since women tend to make greater use of social media and social contact as a means of coping with difficulties, isolation and social distancing are possible negative factors affecting their academic achievement. In addition, female students with caregiving responsibilities for children and parents may have had less time to dedicate to academic work [36,37,38] during stressful conditions while coping with COVID-19. Given that education is a path for promoting gender equality, it is important to investigate the impact of educational institution closure and social isolation among women, especially those engaged in the “help” professions. 

### 1.2. Substance Use

Age is a key factor associated with harmful substance use and other risky behavior [39,40]. Substance use often commences in adolescence and peaks in the first half of emerging adulthood [41]. In Malta, following the first few months of COVID-19, Bonnici et al. (2020) [10] found that study participants who reported substance use before the pandemic claimed increased use of cigarettes (39.7%), alcohol (29.9%), and cannabis (46.9%) because of pandemic fear. Additional results evidence an 8.7% increase in binge drinking, defined as a pattern of alcohol use that corresponds to consuming four or more drinks (female) or five or more drinks (male) in about 2 h [42]. However, over time, the increased level of illicit and licit substance use retreated. A study assessing COVID-19 fear, mental health, and substance misuse among Israeli and Russian social work students, mostly female, shows an increase in substance use during COVID-19. In addition, it was found that respondents who used substances prior to COVID-19 reported more than usual use during the quarantine and social distancing period [29]. Pocuca et al. (2022) [43] found that young adults who lost their jobs felt lonelier, had greater financial concerns, and increased substance use in response to COVID-19. Prowse et al. (2021) [35] found that both males and females used substances to cope during the pandemic and that cannabis use was more likely to have a negative impact on their academic performance and mental health. Forms of stress encountered by university students such as performance pressure, post-graduation planning, financial difficulties, and interpersonal relationships [44] have been found associated with increased substance use [45,46,47]. The gender gap associated with substance use is well documented, with females reporting lower levels except for the nonmedical use of prescription drugs [48] such as amphetamines [41].

### 1.3. Resilience

Reports evidence the impact of COVID-19 on the lives of young people [49] resulting in mental health difficulties [50,51]. The ability to negotiate such an impact and bounce back, resilience, has been examined by Masten et al. (1990) [52] and others. In terms of COVID-19, Peyer et al. (2022) [53] found higher stress levels among female students. Since females tend to rely on social support to cope with stress and share their concerns with others [54,55], as reported above, social distancing and online learning resulting from COVID-19 may have more seriously impacted their ability to be resilient during the pandemic. Bonnici et al. (2020), Yehudai et al. (2020) [10,29], and others note that gender differences in relation to negative student emotions associated with the pandemic should alert researchers and practitioners to young female vulnerability. 

### 1.4. Burnout

Burnout in relation to higher education refers to “a syndrome associated to overwhelming stress caused by a student’s chronic overexposure to excessively pressing demands that he/she is no longer able to meet” [56] (p. 770). COVID-19 placed additional demands and restrictions on student life, resulting in maladaptive emotional and physiological responses [9]. Prolonged stress associated with COVID-19, as well as limited social contact due to lockdowns, tended to be associated with burnout [57]. Among the negative consequences of student burnout are poor academic performance, reduced self-efficacy [58,59], and increased detachment from educational responsibilities and activities [60]. Furthermore, burnout has been associated with eating disorders [61], especially among females [62]. 

### 1.5. Eating Behavior

Healthy eating behavior during COVID-19 was compromised. An international survey conducted by 35 research organizations from Europe, Western Asia, North Africa, and America found many respondents changed their eating patterns during the pandemic with increased unhealthy food consumption, out-of-control eating, and snacking [63]. Such behavior was most probably a mood-driven consequence of increased boredom associated with the pandemic [64].

In sum, the aim of this cross-sectional study was to examine the impact of COVID-19 fear on the well-being of Israeli and Maltese female “help” profession (e.g., social work and psychology) undergraduate students. The central question of the study is to check what cross-national and behavioral differences in the context of the COVID pandemic may be associated with fear of COVID-19. This cross-national comparison includes factors of depression, anxiety, anger, loneliness, nervousness, substance use, eating behavior, burnout, and resilience. The study hypothesis is that country status, including religiosity, is not a significant factor associated with COVID-19 fear impact on select behavioral characteristics of female university students. 

## 2. Methods

Established in 1996, the Ben Gurion University of the Negev Regional Alcohol and Drug Abuse Research (RADAR) Center has received recognition and an award from the US National Institute on Drug Abuse for its contributions to scientific diplomacy through efforts in international collaborative research. For this study, the RADAR Center partnered with University of Malta researchers to collect data and generate usable knowledge about the health and well-being of Israeli and Maltese university students. Specifically, data were collected from “help” profession students about their fear of COVID-19 and its impact on their psycho-emotional well-being. The Qualtrics software platform was used for the survey.

Three scales were used for data collection: (1) the seven-item Fear of COVID-19 Scale (FCV-19S) [25] with two additional questions; (2) the six-item Brief Resilience Scale (BRS) to determine the ability to bounce back or recover from stress [65]; and (3) the ten-item Short Burnout Measure (SBM) to understand a person’s level of physical, emotional, and mental exhaustion [66]. In addition, students were asked about their psycho-emotional well-being during COVID-19, substance use (i.e., tobacco, alcohol, cannabis, or prescription drugs), and how COVID-19 changed their eating behavior.

The survey instruments used were prepared in English and translated into Hebrew; then, they were back translated to English by a team of researchers to ensure content and vocabulary were appropriate. The Cronbach reliability scores of the survey instruments used were FCV-19S = 0.833, BRS = 0.849, and SBM = 0.883. In addition, the Israeli and Maltese investigators received approval from the ethics committees of the universities involved to ensure appropriate steps were taken to protect the rights and welfare of the survey participants. These ethics approval processes are equivalent to established regulations to help protect the rights and welfare of human research subjects [67]. No external grant funding was received for the study. Through local university networks, students of the respective faculties/departments were sent invitations to participate in the survey and sent a link to it. Respondents were advised that the survey was compliant with all ethical standards and that their responses would be confidential and constituted consent to participate.

### 2.1. Statistical Analysis

All statistical analyses were conducted using SPSS, version 25. The Pearson’s chi-squared test for qualitative/nonparametric variables, *t*-test, correlation analysis, and two-way ANOVA and size effect for continuous variables were used. For *t*-test and two-way ANOVA, effect sizes (Cohen’s d and η^2^) were calculated. Stepwise regression analysis was used to identify key COVID-19 fear predictors among the Israeli and Maltese “help” profession students. 

### 2.2. Participants

A total of 453 female “help” profession (i.e., social work and psychology) students completed the online survey from January to July 2021. Respondent details are as follows: 40.4% (n = 184) Israeli and 59.6% (n = 271) Maltese; 50.3% secular and 49.7% non-secular. Table 1 provides information about respondent demographic characteristics.

## 3. Results

The mean ages of the respondents were 26.0 years (SD = 10.8) and 24.8 (SD = 2.6) among the Maltese and Israeli students, respectively (t(451) = 1.565; *p* = 0.118). The mean COVID-19 fear (FCV-19S) scores were similar among the students (21.5 (SD = 5.9) vs. 20.7 (SD = 6.0); t(413) = 1.361; *p* = 0.174). No significant differences were found among the respondents for fear of COVID-19 based on religiosity (t(412) = 0.107; *p* = 0.915). However, two-way ANOVA results evidence a significant difference in COVID-19 fear values based on student country and religiosity statuses (F(1, 410) = 3.940; *p* = 0.048; η^2^ = 0.010) (Figure 1).

Resilience (i.e., BRS score) was found to be higher among Israeli students (20.1 (SD = 4.2) vs. 17.7 (SD = 4.8); t(380) = 5.130; *p* < 0.001; d = 0.536). No significant differences were found in resilience scores based on religiosity (t(379) = 0.950; *p* = 0.343). Two-way ANOVA results evidence no significant difference in resilience values based on country and religiosity (F(1, 377) = 0.034; *p* = 0.853) statuses. 

A higher level of burnout (i.e., SBM score) was reported among Maltese students (27.9 (SD = 8.4) vs. 26.2 (SD = 7.9); t(371) = 1.991; *p* = 0.047; d = 0.210). No significant differences were found based on religiosity (t(370) = 0.598; *p* = 0.550). Two-way ANOVA results evidence no significant difference in burnout values based on country and religiosity (F(1, 368) = 3.454; *p* = 0.064) statuses.

Substance use (i.e., tobacco, alcohol, cannabis, stimulants, and/or prescription drugs) in the last month was reported by 77.2% of the respondents. No significant differences were found for previous-month substance use based on country status (χ^2^(1) = 2.495, *p* = 0.114). Israeli students reported more alcohol (78.9% vs. 68.0%; χ^2^(1) = 5.524, *p* = 0.019), cannabis (26.0% vs. 11.0%; χ^2^(1) = 14.664, *p* < 0.001), and stimulant use (16.9% vs. 3.1%; χ^2^(1) = 21.950, *p* < 0.001). However, Maltese students reported a higher level of previous-month binge drinking (15.6% vs. 6.2%; χ^2^(1) = 8.161, *p* = 0.004). No significant differences were found for previous-month pain reliever and sedative use based on country status (χ^2^(1) = 0.096, *p* = 0.757, and χ^2^(1) = 1.730, *p* = 0.188, respectively). Regardless of country, secular students reported more substance use than non-secular students (86.2% vs. 67.6%; χ^2^(1) = 18.281, *p* < 0.001). Two-way ANOVA results evidence a significant difference in COVID-19 fear values based on country and previous-month substance use (F(1, 358) = 5.031; *p* = 0.026; η^2^ = 0.014) (Figure 2).

Due to COVID-19, 23.9% of the respondents reported an increase in substance use in the last month. However, no significant differences were found for previous-month substance use increase based on country status (χ^2^(1) = 1.189, *p* = 0.276). Regardless of country, respondents who reported more previous-month substance use had higher COVID-19 fear (t(374) = 3.419; *p* < 0.001; d = 0.415) and burnout (t(364) = 3.888; *p* < 0.001; d = 0.477) scores, as well as lower resilience scores (t(373) = 2.321; *p* = 0.018; d = 0.287).

Due to COVID-19, most respondents (74.3%) reported deterioration of their psycho-emotional well-being in the last month with complaints of depression (36.5%), exhaustion (56.4%), loneliness (47.0%), nervousness (49.2%), and anger (41.8%). No significant differences were found for deterioration of psycho-emotional well-being in the last month based on country and religiosity statuses (χ^2^(1) = 0.022, *p* = 0.882, and χ^2^(1) = 1.837, *p* = 0.175, respectively). Regardless of country status, respondents who reported deterioration of their psycho-emotional well-being in the last month had more fear of COVID-19 (t(373) = 8.812; *p* < 0.001; d = 1.054) and burnout (t(364) = 8.419; *p* < 0.001; d = 1.014), as well as lower resilience scores (t(373) = 4.720; *p* < 0.001; d = 0.560). In addition, respondents who reported deterioration of their psycho-emotional well-being reported more previous-month substance use (27.9% vs. 12.2%; χ^2^(1) = 9.754, *p* = 0.002).

Among all respondents, 68.4% reported eating more salt and sugar-loaded food because of COVID-19. No significant differences were found for eating behavior changes based on country and religiosity statuses (χ^2^(1) = 3.806, *p* = *0*.051, and χ^2^(1) = 0.663, *p* = 0.415, respectively). Additionally, 49.2% of the respondents reported weight increase. No significant differences were found for weight increase based on country and religiosity statuses (χ^2^(1) = 0.028, *p* = 0.868, and χ^2^(1) = 0.123, *p* = 0.726, respectively).

Regardless of country, respondents who reported eating more salt and sugar-loaded food because of COVID-19 had more fear of infection (t(356) = 3.410; *p* < 0.001; d = 0.388) and burnout (t(361) = 3.119; *p* = 0.002; d = 0.320), as well as lower resilience (t(362) = 2.833; *p* = 0.005; d = 0.320). In addition, respondents who experienced eating behavior changes during COVID-19, reported more previous-month substance use (28.3% vs. 12.4%; χ^2^(1) = 11.027, *p* = 0.001) and psycho-emotional well-being deterioration (80.4% vs. 62.6%; χ^2^(1) = 13.128, *p* < 0.001).

Israeli and Maltese student fear of COVID-19 was examined as a dependent variable for stepwise regression analysis. Two significant predictors were found explaining COVID-19 fear—burnout and psycho-emotional well-being deterioration. Other independent variables (i.e., country, age, religiosity, resilience, and substance use) did not provide a significant increment in the proportion of variance explained. The resulting value of the explained variance (adjusted R^2^) for the dependent variable (fear of COVID-19) was 0.171 (17.1%). Results of regression analysis for each country evidence the same significant predictors of COVID-19 fear—burnout and psycho-emotional well-being deterioration. The values of the explained variance (adjusted R^2^) for the dependent variable (fear of COVID-19) were 0.167 and 0.184 for Israel and Malta, respectively. For both countries, increased student burnout and deterioration in student psycho-emotional well-being due to COVID-19 tend to increase student fear of infection. Regardless of the country, a positive correlation was found between COVID-19 fear and burnout (r = 0.328; *p* < 0.001) as well as between COVID-19 fear and deterioration in psycho-emotional well-being (r = 0.404; *p* < 0.001).

## 4. Discussion and Conclusions

This cross-sectional, comparative study examined the impact of COVID-19 fear on the well-being of Israeli and Maltese female “help” profession (i.e., social work and psychology) undergraduate students. Such students are or will be on the front line of providing health and social services to people in need.

Our findings evidence similar levels of COVID-19 fear among Israeli and Maltese students. Regression analysis shows that fear-related predictors are the same for both countries. Resilience was found to be higher among Israeli students, and burnout was found to be more prevalent among those from Malta. Substance use (i.e., tobacco, alcohol, cannabis, stimulants, or prescription drugs) in the last month was reported by 77.2% of the respondents. No significant differences were found for previous-month substance use based on country status. However, respondents who reported more previous-month substance use had higher COVID-19 fear and burnout scores, as well as lower resilience scores. This should be considered when planning and developing substance use prevention measures for students. Due to COVID-19, most respondents (74.3%) reported deterioration of their psycho-emotional well-being in the last month; however, no differences were found based on country and religiosity statuses. In addition, no significant differences were found for eating behavior changes and weight increase based on country and religiosity statuses.

The results obtained tend to confirm the study hypothesis that country status, including religion, is not a significant factor associated with COVID-19 fear impact on select behavioral characteristics of female university students. Most behavioral characteristics (i.e., fear, burnout, substance use, and eating behavior), except for resilience, were similar between Israeli and Maltese students.

Among the study limitations, this cross-sectional survey is based on a limited cross-sectional cohort of students from two countries at approximately the same time of the COVID-19 pandemic. An additional study over time involving other countries is needed to better understand the short- and long-term impact of COVID-19. It will also be useful to consider cross-cultural differences between the two countries, including the quality of life and opportunities for self-realization. Furthermore, this study examined only female students; additional research is needed regarding the impact of COVID-19 fear on the well-being of male students to further understand and investigate possible gender differences.

In conclusion, this study shows that COVID-19 fear had an impact on Israeli and Maltese female “help” profession students, affecting their well-being from multiple perspectives. Therefore, prevention and intervention measures, including those that can be made available on campus, should be considered by university administration personnel and student association leaders in consultation with mental health professionals.

## Figures and Tables

**Figure 1 ijerph-20-03968-f001:**
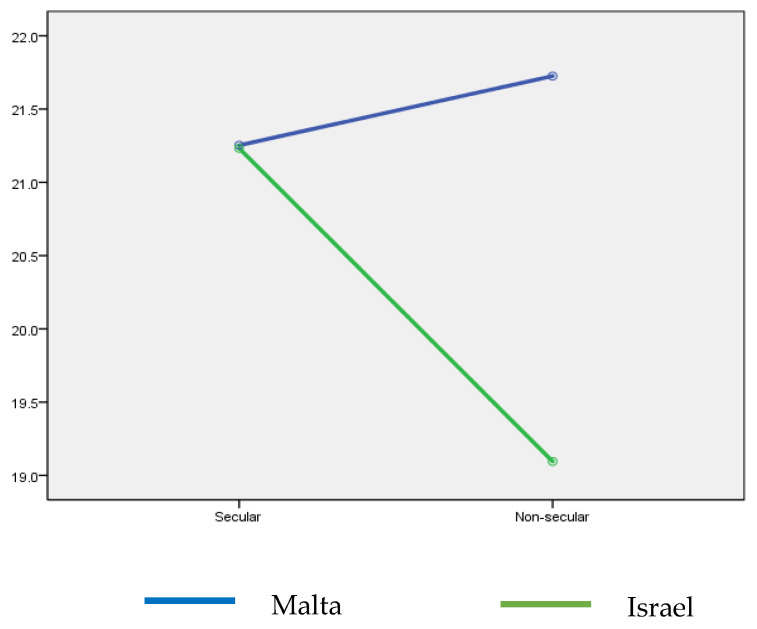
Fear of COVID-19 by country and religiosity.

**Figure 2 ijerph-20-03968-f002:**
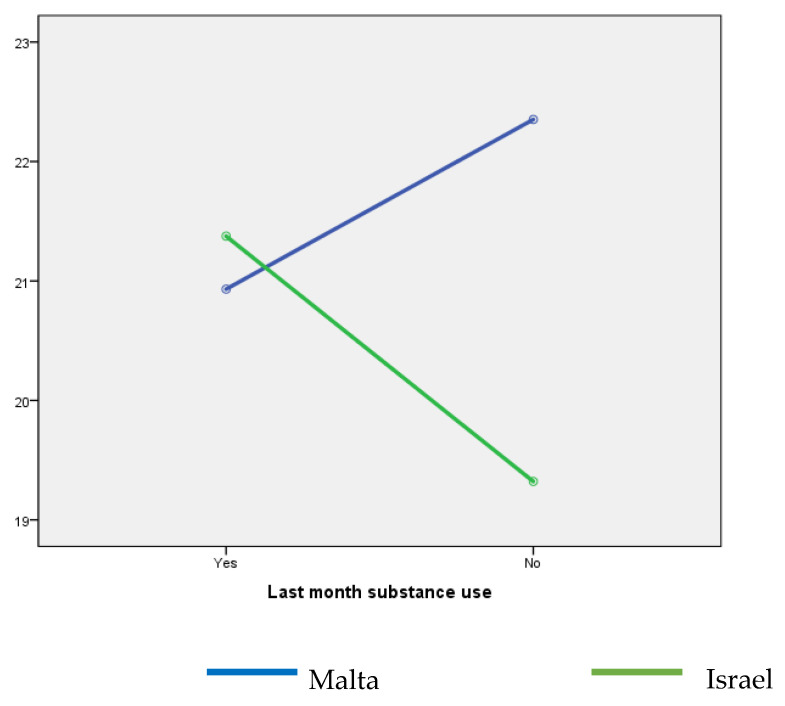
Fear of COVID-19 by country and previous-month substance use.

**Table 1 ijerph-20-03968-t001:** Demographic characteristics.

	Total (n = 455) ^1^	Israel (n = 184)	Malta (n = 271)	*p* Value
Age, Mean (SD) Median	25.5 (8.5) (24.0)	24.8 (2.6) (25.0)	26.0 (10.8) (21.0)	0.118
Religiosity, % (n) Secular Non-secular	50.3 (228) 49.7 (225)	74.9 (137) 25.1 (46)	33.7 (91) 66.3 (179)	<0.001

^1^ Up to two subjects in each group missing data on some variables.

## Data Availability

The data presented in this study are available on request from the corresponding author. The data are not publicly available due to local restrictions.

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
