# Peer review of "COVID-19 Fear Impact on Israeli and Maltese Female “Help” Profession Students"

_ijerph, 2023, doi:10.3390/ijerph20053968_

Round 1

Reviewer 1 Report

The subject of the article is very interesting and written well. However, some issues need to be addressed and are presented in the comments indicated in the body of the attached article. 

Author Response

Reviewer 1 - Co-authors and I appreciate the helpful suggestions provided.  Revisions have been made to the paper addressing the wording/reference on line 100, correction of reference number on line 113; correction of the italicized paragraph to normal font; and discussion added about the meaning of the findings.  

Again, thanks you for your cooperation.

Reviewer 2 Report

The study is interesting; there is some below suggestion,

  1.      Page 5, “Table 1. Demographic Characteristics,” presented Total (n=455), Secular is 228, and Non-secular is 225, the account is 453; in addition, Malta , and Israel have the same problem, please clarify.

2.         Most sections, show “Error! Reference source not found.].” please carefully revise.

3.         On page 4, for more contributions, suggest authors add the research question after the study’s aim.

4.          Methods section, how to collect Participants is unclear.

5.         Discussion and Conclusion section, suggest responses and discuss more via “The study hypothesis is that country status, including religiosity, is not a significant factor associated with COVID-19 fear impact on select behavioral characteristics of female university students” for example, what kind of the behavioral characteristics of female university students would be?

Author Response

Coauthors and I appreciate the reviewer's helpful comments. 

Reviewer 3 Report

The article is devoted to the study of an important issue in the cross-cultural aspect. The introduction contains all the necessary information about previous studies. The authors present the results of the study using adequate methods of analysis.

At the same time, there are some positions in the article that need clarification.

The absence of tables in the text with the average indicators of respondents in the two countries and indicators of the criterion of the significance of differences makes the text somewhat less perceptible.

The authors do not explain (do not interpret) the lower rates of resilience and higher rates of fear of covid 19 and burnout in Maltese women. Doesn't this seem to be related to the differences in living conditions in the two countries: it's not even about the socio-economic level or the level of development of medicine, but significant differences, for example, opportunities for self-realization. Moreover, these are important differences in connection with the results of regression analysis, since these variables act as predictors of fear.

There are technical errors in the text in the form of errors instead of references (lines 43, 58, 61, etc.).

It would be desirable to reflect the results of regression analysis in the conclusions and maybe look separately by country: are there correlations or regressions. In case of absence, this also needs to be explained.

Author Response

Co-authors and I appreciate the reviewer's helpful comments.
